# Association of Maternal Inactivated COVID-19 Vaccination within 3 Months before Conception with Neonatal Outcomes

**DOI:** 10.3390/vaccines11111710

**Published:** 2023-11-10

**Authors:** Zhihui Chen, Xingbo Mu, Xinyan Wang, Lulu Zhang, Ge Liu, Jun Zheng, Fangrui Ding

**Affiliations:** 1Department of Neonatology, Tianjin Central Hospital of Obstetrics and Gynecology, Tianjin 300100, China; 13845104823@163.com (Z.C.); muxingbo1984@126.com (X.M.); lulubjmu1915@163.com (L.Z.); happy.lg@163.com (G.L.); 2Tianjin Key Laboratory of Human Development and Reproductive Regulation, Tianjin 300199, China; 5020210003@nankai.edu.cn; 3Department of Neonatology, Nankai University Maternity Hospital, Tianjin 300052, China; 4Center for Reproductive Medicine, Tianjin Central Hospital of Obstetrics and Gynecology, Tianjin 300100, China

**Keywords:** COVID-19, vaccination, preterm birth, small for gestational age, NICU admission

## Abstract

There is limited available data addressing whether inactivated COVID-19 vaccination before conception is associated with adverse neonatal outcomes. This cohort study included all singleton live births at our center from March 1 to June 30, 2022. According to whether a maternal inactivated COVID-19 vaccination had been administered within 3 months before conception or not, neonates were identified as being in the vaccinated or unvaccinated group. Vaccination information and clinical characteristics were extracted for analysis. Furthermore, neonatal outcomes were analyzed and compared between these two groups in the present study. The cohort included 856 eligible newborns, of whom 369 were exposed to maternal vaccination before conception and 487 were unexposed newborns. No differences were observed in rates of preterm birth, newborns being small for gestational age, or neonatal intensive care unit admission between exposed and unexposed newborns. Furthermore, even after adjusting for social–economic status and maternal characteristics, there remained no significant differences in these neonatal outcomes. Our study revealed no statistically significant differences between newborns born to women who received inactivated vaccines prior to conception compared with those who did not receive any vaccinations. In addition, our study also highlights the importance of considering COVID-19 vaccination before conception.

## 1. Introduction

On 5 May 2023, the Director of the WHO declared Coronavirus disease 2019 (COVID-19) to no longer be a global health emergency. This suggests that we have stepped into a new stage in preventing and treating COVID-19 [1]. However, according to the WHO COVID-19 Weekly Epidemiological Update, over 868,000 new COVID-19 cases and over 3700 fatalities were reported from 26 June to 23 July 2023 [2]. These numbers are likely underestimated due to reduced testing and reporting globally. As such, COVID-19 remains one of the most pressing global health concerns. Since the first COVID-19 vaccination was initiated at the end of 2020, the effectiveness of vaccination against COVID-19 has been reported by many studies [3,4,5,6,7,8,9,10]. During the COVID-19 pandemic, COVID-19 vaccination offered protection against infection and significantly reduced hospital admissions and mortality rates associated with COVID-19 [3,4,5,6,7,8,9,10]. This holds true for special populations, including pregnant women, as vaccination effectively prevents severe cases of COVID-19 and decreases morbidity and mortality rates in this group. COVID-19 vaccination is still one of the most effective ways of preventing COVID-19 [3,4,5,6,7,8,9,10]. Nevertheless, there remain several issues surrounding vaccination against COVID-19 in special populations such as pregnant women or those preparing for pregnancy.

Available data have already shown that COVID-19 during pregnancy is associated with COVID-19-related morbidity and mortality [11,12,13,14]. Furthermore, fetuses and neonates are also at increased risk of adverse outcomes in COVID-19-infected maternal individuals [11,12,13,14]. Nowadays, numerous studies have provided evidence showing that vaccination against COVID-19 (primarily mRNA vaccines) during pregnancy does not pose an increased risk of adverse outcomes for pregnant women or their offspring [6,7,8,9,10]. Based on these findings, the WHO, government agencies, and perinatal associations recommend administering the COVID-19 vaccine to pregnant individuals [7,15,16,17]. Currently, due to the evolving COVID-19 pandemic, pregnant women may no longer need to decide whether to proceed with vaccination against COVID-19 during pregnancy [1]. However, a crucial concern remains regarding the impact of COVID-19 vaccination during preconception preparation on pregnancy and neonatal health. Additionally, it is essential to determine the optimal time period for vaccinated women who are preparing for pregnancy to begin attempting conception. These pressing issues necessitate immediate attention. Although most government agencies and perinatal associations recommend that individuals who are planning for pregnancy should get vaccinated as soon as possible [15,16,18], there is limited data addressing whether COVID-19 vaccination prior to conception could impact pregnancy outcomes.

Currently, most of the available information regarding the safety of COVID-19 vaccination during pregnancy is based on mRNA vaccines [7,8], with limited data on inactivated COVID-19 vaccines [16,19]. Furthermore, while maternal outcomes have been extensively studied, neonatal outcomes remain understudied. To provide evidence-based suggestions, we collected information from women who received an inactivated COVID-19 vaccine within three months prior to conception and examined its association with neonatal outcomes.

## 2. Materials and Methods

This study was approved by our hospital Institutional Review Board (No. 2022KY049). All study protocols adhered to the Declaration of Helsinki and written informed consent was obtained from all parents of the participants prior to their inclusion in this study. The study followed the Strengthening the Reporting of Observational Studies in Epidemiology (STROBE) reporting guidelines.

### 2.1. Study Population

This cohort study included all singleton live births delivered at Tianjin Central Hospital of Obstetrics and Gynecology in China between 1 March 2022 and 30 June 2022. Our hospital is the largest Obstetrics and Gynecology Hospital in Tianjin, with more than 12,000,000 people in this city. Nearly one-fifth of all newborns in this city are delivered in our hospital annually. We provide comprehensive care for both high-risk and low-risk pregnant women, who come from various districts within our city. Moreover, these pregnant women share similar sociodemographic characteristics with the overall pregnant population in our city. Newborns whose mothers received the first dose of the COVID-19 vaccine before conception were identified as the vaccinated group. Newborns whose mothers had not been vaccinated before or during pregnancy were assigned to the unvaccinated group. The collection of COVID-19 infection information was conducted by questionnaire, and it was confirmed that none of the newborns’ mothers had been infected with COVID-19 either before or during pregnancy. Gestational age was determined using either the last menstrual period (LMP) or ultrasound measurements. The number of days from pre-pregnancy was calculated on the basis of the LMP. Vaccination information such as date of COVID-19 vaccination and type of COVID-19 vaccine were obtained from the Chinese National Government Service Platform. The clinical characteristics of both pregnant women and newborns were extracted from electronic medical records.

The exclusion criteria were as follows: (1) failure to obtain informed consent; (2) non-singleton births, due to this being closely related to outcomes of preterm birth and low birth weight; (3) invalid or incomplete COVID-19 vaccine information; (4) receipt of an untargeted COVID-19 vaccine; (5) incomplete clinical information; (6) time of vaccination against COVID-19 more than 3 months before conception (90 days); (7) Any dose of vaccination against COVID-19 during pregnancy.

### 2.2. Primary Outcomes

The primary outcomes of this study were proportions of preterm births, newborns who were small for gestational age (SGA) among live births, and admission to the neonatal intensive care unit (NICU). Preterm birth was defined as birth before the 37th gestational week. SGA was defined as a birth weight less than the 10th percentile for gestational age at birth. The proportion of NICU admissions was defined as percentage of neonates admitted to the NICU among all the included neonates in the different groups.

### 2.3. Secondary Outcomes

The secondary outcomes included very preterm birth (<32 weeks), late preterm birth (32–37 weeks), low birth weight (<2500 g), very low birth weight (<1500 g), neonatal bacterial infection, and mortality. Very preterm birth was defined as a gestational week of less than 32 weeks. Late preterm birth was defined as a gestational week of between 32 and 37 weeks. Low birth weight was identified as a birth weight of less than 2500 g, while a very low birth weight was less than 1500 g. Neonatal bacterial infection was defined as the recovery of a bacterial pathogen from the blood or cerebrospinal fluid during hospitalization, or as testing negative for bacterial culture whilst having abnormal infectious indicators and clinical features. Mortality was defined as death before discharge. Infants transferred to another hospital were tracked for survival status until discharge from the hospital.

### 2.4. Statistical Analysis

The demographic and clinical characteristics of the mother–infant cohort were described according to the COVID-19 vaccination maternal status. We used the Shapiro–Wilk test and quantile–quantile (Q–Q) and probability–probability (P–P) plots to assess the normality of continuous variables. Differences in means or medians between groups were compared using Student’s *t* test or the Mann–Whitney U test for continuous variables, and differences in proportions were compared using the Chi-square or Fisher’s exact test for categorical variables. Odds ratios (ORs) and corresponding 95% confidential intervals (CIs) for all dichotomous outcomes were obtained using logistic regression. Covariates in multivariable models were selected with reference to previous studies, clinical relevance, and relation to vaccination status and obstetric outcomes. We developed four sets of models: (1) Model 1: crude model; (2) Model 2: adjusted for social–economic status, including per capita annual disposable income levels (<4380 United States dollar (USD) per year vs. ≥4380 USD per year, according to the Statistical Bulletin of the People’s Republic of China on National Economic and Social Development in 2020) and highest level of education received (junior high or below, senior high, college or above); (3) Model 3: further adjusted for maternal age at delivery, obesity during pregnancy, whether the conception was planned or not and whether it was achieved through assisted reproductive technology, history of abnormal pregnancy, and whether there was a previous preterm infant. The main analysis evaluated any vaccination during the 3 months prior to the last menstrual period, while secondary analyses examined differences between vaccination 2 or 3 months prior to the last menstrual period. In addition, the probability of primary neonatal outcomes was plotted against the time between the first dose of vaccination and the LMP.

Analyses were performed with the use of SAS software, version 9.4 (SAS Institute Inc., Cary, NC, USA). All tests were two-sided, and *p* < 0.05 was considered statistically significant.

## 3. Results

A total of 2606 infants were born at our center from March to June; 1750 were excluded because informed consent was unavailable or due to multiple births, unavailable vaccination information, non-inactivated COVID-19, unavailable clinical information, vaccinations received over 3 months before conception, and vaccination during pregnancy (Figure 1). After exclusion, a total of 856 neonates remained eligible for analysis, of whom 369 were exposed to maternal vaccination before conception.

### 3.1. Demographic and Baseline Characteristics

As shown in Table 1, the vaccinated group was characterized by higher education levels compared to the unvaccinated group. In terms of clinical maternal characteristics, the rate of unplanned pregnancy was higher in vaccinated group. Additionally, a lower rate of history of abnormal pregnancy and a lower rate of assisted reproductive technology use were identified in this group.

### 3.2. Primary Outcomes

As show in Table 2, there were no differences between the vaccinated and unvaccinated groups in the proportion of newborns with preterm birth, SGA, and NICU admission. As shown in Figure 2, when adjusted by social–economic status, there was no difference between the two groups in terms of neonatal outcomes in preterm birth, SGA, and NICU admissions. After further adjustment for maternal characteristics, there were still no differences between the two groups.

### 3.3. Secondary Outcomes

In Table 2, more detailed neonatal outcomes are listed. No significant difference was observed in terms of very preterm birth (<32 weeks), late preterm birth (32–37 weeks), low birth weight (<2500 g), very low birth weight (<1500 g), or neonatal bacterial infection. As for mortality, no cases of death were found in any of these births.

### 3.4. Subgroup Analysis According to the Timing of COVID-19 Vaccination

Considering that different timings of COVID-19 vaccination before conception may contribute to different immune responses, a subgroup analysis was performed based on time intervals from initial vaccination to conception. Vaccinated cases were further subdivided into two subgroups: within two months (60 days or less) and within three months (61 to 90 days). The number of newborns in the two subgroups included in the study were, respectively, 184 and 185 (Table 3). When compared with the unvaccinated group, no difference was observed in any of the subgroups. To clearly present this result, the probability of all three primary neonatal outcomes was analyzed according to vaccination time intervals, included as a continuous variable. The probability rate of preterm births seemed to be the highest when the time interval (days) was 0, and then the rate linearly decreased with increasing time intervals (Figure 3). The probability rate of SGA and NICU admission appeared to be level over time (Figure 3).

### 3.5. Subgroup Analysis According to Different Vaccine Types

There are two different inactivated COVID-19 vaccine types: Sinopharm-BIBP and Sinovac-Coronavac. Different vaccine types may contribute to different results. A subgroup analysis based on different vaccines types was applied to compare the unvaccinated group with different vaccines. As shown in Table 4, there were no statistical difference in these subgroups.

## 4. Discussion

This study detected no increased risk of adverse neonatal outcomes such as preterm birth, SGA, or NICU admission among live-born infants whose mothers were exposed to inactivated COVID-19 vaccines before conception compared to unexposed individuals. After adjustment for social–economic status and maternal characteristics, there were no statistically significant differences between the two groups. In addition, when the vaccinated group was sub-grouped based on the time interval between initial vaccination and conception and by different vaccine types, no increased risk for negative neonatal outcomes was found in any of the subgroups.

The majority of studies have shown that vaccination for COVID-19 (primarily mRNA vaccines) during pregnancy does not increase the risk of adverse neonatal outcomes [6,7,8]. However, given the current situation with the ongoing pandemic, there is limited data available to address concerns regarding the optimal timing for conception after COVID-19 vaccination. Women planning a pregnancy may be more concerned about this issue than with which trimester of pregnancy it is safe to receive vaccination. In China, according to rigorous policy, there were a limited number of COVID-19 infection cases that occurred from March 2020 to November 2022. Furthermore, after the end of March 2021, large numbers of people received inactivated COVID-19 vaccines [20]. Although pregnant women were excluded at the initial stages of pregnancy at that time, a number of women of childbearing age were vaccinated before conceiving. Thus, these cases provided a good opportunity to explore the effects of vaccination against COVID-19 before conception on their offspring. In this study, we examined the neonatal outcomes of mothers who were vaccinated within 3 months before conception (Table 2 and Figure 2). No increase in adverse neonatal outcomes was observed between the exposed and unexposed groups.

The most widely used COVID-19 vaccines are mRNA and inactivated vaccines [17,21,22]. Numerous studies have demonstrated that mRNA COVID-19 vaccines do not increase adverse neonatal outcomes [6,7,8,9]. However, the majority of these studies related to mRNA COVID-19 vaccines are focused on maternal vaccination during pregnancy instead of before conception. In December 2022, Li et al. reported on the safety of maternal inactivated COVID-19 vaccines before conception in terms of maternal and neonatal outcomes [23]. However, certain neonatal outcomes such as preterm birth, SGA, and low birth weight were not included in their report. Thus, the present study provides data on inactivated vaccines in relation to these neonatal outcomes. In addition, considering the rapid advancements in vaccine technology and the global trend towards the use of mRNA vaccines, whether mRNA COVID-19 vaccination before conception is associated with adverse neonatal outcomes could also be examined in future studies.

As shown in Figure 2, as one of the included neonatal outcomes, the proportion of SGA in the vaccinated group was higher but not statistically different. In addition, regarding SGA, in the subgroup analysis investigating the timing of COVID-19 vaccination, the values of OR were especially high in the subgroup who received a vaccination between 2 and 3 months prior to conception, after adjustment—as shown in Table 3. While limited data exist regarding the effects of pre-conception COVID-19 vaccination on SGA infants specifically, several studies investigating vaccination (primarily mRNA vaccines) during pregnancy have reported no differences between vaccinated and unvaccinated groups with respect to SGA infants [7,8,9]. Caution should be exercised when interpreting the results of this study due to limitations such as the non-population-based design, the relatively small sample size (the number of SGA in the subgroup was small and the upper 95%Cl was large), and the different types of COVID-19 vaccines; therefore, further studies with larger participant numbers are warranted to confirm these findings. Additionally, these results still suggest to us that the influence on fetuses or newborns caused by vaccination before conception or in the early stages of pregnancy is non-ignorable. Theoretically, the activation of the maternal immune response may potentially influence fetal acceptance, placental development, and the development of preeclampsia and SGA in newborns [24]. It is unfortunate that the present study did not examine serum antibody levels of individuals vaccinated against COVID-19; otherwise, it could have potentially provided valuable support for this deduction.

During the collection of baseline information, we conducted a rough survey in the unexposed group to explore their hesitancy towards accepting COVID-19 vaccination. The results showed that the most common concern was the safety of the vaccine—particularly among women with a history of abnormal pregnancies or those undergoing assisted reproduction during pregnancy. These individuals were extremely cautious when planning for pregnancy. This observation is indirectly supported by a lower rate of unplanned pregnancies in the unvaccinated group and a higher utilization rate of assisted reproductive technology compared to the vaccinated group. Thus, we have adjusted these factors in our analysis. As show in Figure 2, no adverse risk was examined in the vaccinated group after adjustment. This issue has been also paid attention to in the fields of reproduction and embryology [25]. Most studies investigating COVID-19 vaccination related to IVF treatment have not identified any adverse pregnancy outcomes when comparing women who receive COVID-19 vaccines with those who remained unvaccinated [26,27,28]. However, as showed in the Shi et al. study, a significantly reduced pregnancy rate was observed among patients who received their first dose of the COVID-19 vaccine 30 days or less or 31 to 60 days before IVF treatment, and a slightly but not statistically lower rate of pregnancy was found in the 61 to 90 days’ subgroup [29]. Thus, we have identified the vaccinated group as individuals who received the vaccine within three months before conception. In addition, subgroup analyses based on the timing of COVID-19 vaccination were also performed. As presented in Table 3, there were no significant differences observed between the vaccinated and unvaccinated groups regarding neonatal outcomes across different time intervals before conception. However, when presenting the probability of preterm birth according to time intervals, as a continuous variable, the probability rate of preterm birth seemed to be the highest when the time interval (days) was 0, and then the rate linearly decreased with increasing time intervals (Figure 3). This result also suggested to us that the influence on fetuses or newborns caused by vaccination before conception or in the early stages of pregnancy is non-ignorable.

Several limitations must be acknowledged when interpreting our findings: Firstly, this study was an observational study, not a randomized controlled trial. Secondly, all cases included in this study were from a single city in China, and so the study was not population-based. This may limit the generalizability of the study’s findings. Moreover, the duration of this study, which encompassed cases occurring between March and June, was short. To mitigate the potential impact of varying temperatures and humidity on the results, it would be more advantageous to extend the study period to one year. Additionally, due to limited numbers of individuals who had only received one vaccination dose, we could not conduct a detailed analysis based on different dosages.

## 5. Conclusions

Our study highlights the importance of considering COVID-19 vaccination before conception and found no significant differences between newborns born to women who received inactivated vaccines prior to conception compared with those who did not receive any vaccinations. Thus, our data supports the implementation of COVID-19 vaccinations for women preparing for pregnancy. Additionally, it suggests that inactivated COVID-19 vaccinations administered within 3 months before conception appear to be safe for newborns.

## Figures and Tables

**Figure 1 vaccines-11-01710-f001:**
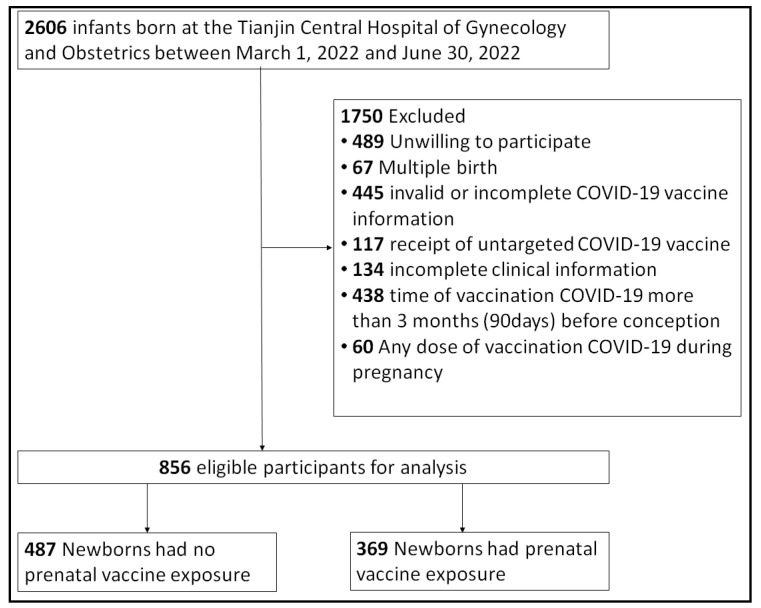
**Flowchart of Newborn Selection for Cohort Study of maternal inactivated COVID-19 vaccination within 3 months before conception**. A total of 2606 infants were born in our center from March to June 2022; 1750 were excluded, and a total of 856 neonates remained eligible for analysis, of whom 369 were exposed to maternal vaccination before conception.

**Figure 2 vaccines-11-01710-f002:**
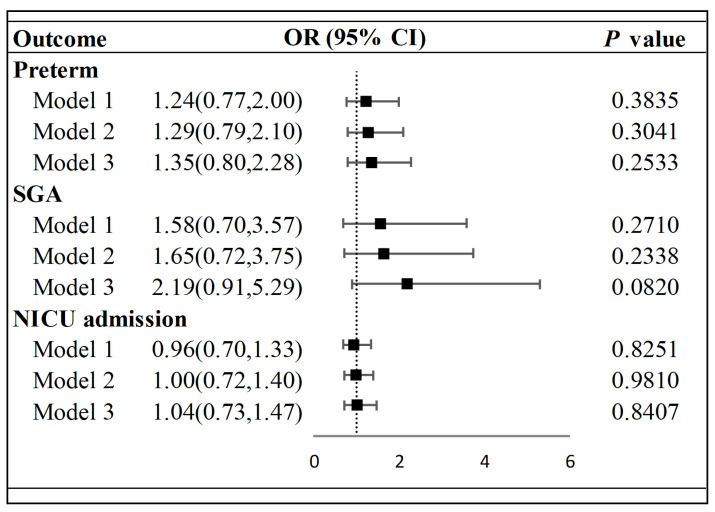
**Primary neonatal outcomes adjusted by different models.** Primary neonatal outcomes including preterm, SGA, and NICU admissions have been adjusted by different models and shown in forest plots. (Model 1: crude model; Model 2: adjusted for social–economic status, including per capita annual disposable income levels and education levels; Model 3: further adjusted for maternal age at delivery, obesity during pregnancy, whether the conception was planned or not and whether it was achieved through assisted reproductive technology, history of abnormal pregnancy, and whether there was a preterm infant.). Abbreviations: SGA, small for gestational age. OR: odds ratio; CI, confidence interval.

**Figure 3 vaccines-11-01710-f003:**
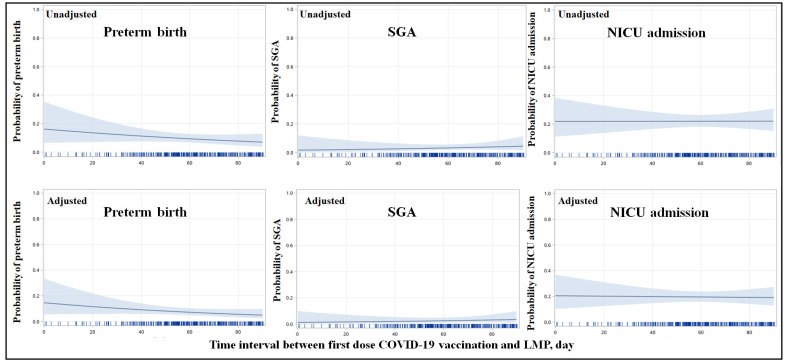
**Probability of neonatal outcomes according to the time interval between the first dose of vaccination and the last menstrual period (LMP), as a continuous variable**. The horizontal axis represents the time interval between the 1st dose of the COVID-19 vaccine and the LMP in days, as a continuous variable, and the vertical axis represents the probability of a given neonatal outcome. The spline smoothing plot and the shaded areas represent the probability of the neonatal outcome and its 95% CI. Covariates in the adjusted model included: per capita annual disposable income (<4380 USD per year vs. ≥4380 USD per year) and education level (junior high or below, senior high, college or above), maternal age at delivery, obesity during pregnancy, whether the conception was planned or not and whether it was achieved through assisted reproductive technology, history of abnormal pregnancy, and whether there was a previous preterm infant.

**Table 1 vaccines-11-01710-t001:** Baseline characteristics of mothers with a singleton live birth according to exposure to inactivated COVID-19 vaccination within 3 months before conception.

	Unvaccinated	Vaccinated	*p* Value
**Demographic characteristics**	N = 487	N = 369	
Maternal age, years	31.49 ± 4.27	31.50 ± 4.26	0.9669
Ethic Han, n (%)	473 (97.13)	362 (98.10)	0.3598
Education, n (%)			0.0485
Senior high or below	143 (29.36)	83 (22.49)	
College or higher	344 (70.64)	286 (77.51)	
Per capita annual disposable income, n (%)			0.1545
<4380 USD	94 (19.30)	86 (23.31)	
≥4380 USD	393 (80.70)	283 (76.69)	
**Vaccination Information**			
Types of inactivated, n (%)			<0.0001
Sinopharm-BIBP	0 (0.00)	231 (62.6)	
Sinovac-Coronavac	0 (0.00)	138 (37.4)	
Dose of maternal vaccination, n (%)			<0.0001
0	487 (100.00)	0 (0.00)	
1	0 (0.00)	42 (11.38)	
2	0 (0.00)	327 (88.62)	
**Preexisting maternal comorbidities**			
DM, n (%)	2 (0.41)	3 (0.81)	1.0000
HTN, n (%)	1 (0.21)	0 (0.00)	0.6568
CVD, n (%)	0 (0.00)	1 (0.27)	0.4311
DM/HTN/CVD, n (%)	3 (0.62)	4 (1.08)	0.4722
History of gestation			
History of abnormal pregnancy, n (%) ^a^	68 (13.96)	33 (8.94)	0.0242
History of preterm delivery, n (%)	1 (0.21)	1 (0.27)	1.0000
Unplanned pregnancy, n (%)	32 (6.57)	92 (24.93)	<0.0001
Ways of getting pregnancy, n (%)			<0.0001
Natural	391 (80.29)	350 (94.85)	
ART	96 (19.71)	19 (5.15)	
GDM, n (%)	112 (23.00)	82 (22.22)	0.7883
HDCP, n (%)	27 (5.54)	27 (7.32)	0.2907
PE, n (%)	20 (4.11)	15 (4.07)	0.9756
Hypothyroidism during pregnancy, n (%)	29 (5.95)	23 (6.23)	0.8660
Abnormality of placenta, n (%) ^b^	42 (8.62)	29 (7.86)	0.6877

Abbreviations: DM, diabetes mellitus; HTN, hypertension; CVD, cardiovascular disease; ART, assisted reproductive technology; GDM, gestational diabetes mellitus; HDCP, hypertensive disorder complicating pregnancy; PE, pre-eclampsia; USD, United States dollar. ^a^: History of abnormal pregnancy is identified as a history of embryonic resorption, fetal abortion, or ectopic pregnancy. ^b^: Abnormality of the placenta is identified as placental morphology abnormalities, placental size abnormalities, and placental position abnormalities such as placenta abruption, placenta previa, velamentous placenta, and so on.

**Table 2 vaccines-11-01710-t002:** Neonatal Characteristics and Outcomes.

	Unvaccinated	Vaccinated	*p* Value
	N = 487	N = 369	
Female, n (%)	245 (50.31)	179 (48.51)	0.6022
Cesarean delivery, n (%)	224 (46.00)	156 (42.28)	0.2781
SGA, n (%)	11 (2.26)	12 (3.25)	0.3734
Preterm birth (<37 weeks), n (%)	38 (7.80)	35 (9.49)	0.3828
Very preterm birth (<32 weeks), n (%)	7 (1.44)	4 (1.08)	0.7653
LBW (<2500 g), n (%)	35 (7.19)	27 (7.32)	0.9420
VLBW (<1500 g), n (%)	6 (1.23)	3 (0.81)	0.7393
NICU admission, n (%)	110 (22.59)	81 (21.95)	0.8248
Neonatal bacterial infection, n (%)	33 (6.78)	22 (5.96)	0.6305
Mortality, n (%)	0	0	1.0000

Abbreviations: SGA, small for gestational age; LBW, low birth weight (<2500 g); VLBW, very low birth weight (<1500 g).

**Table 3 vaccines-11-01710-t003:** Primary neonatal outcomes in the subgroup analysis according to the timing of COVID-19 vaccination.

	Unvaccinated	Within 2 Month	Between 2~3 Month
	N = 487	N = 184	N = 185
**Preterm**	OR (95%CI)	OR (95%CI)	OR (95%CI)
Model 1	1.00 (Reference)	1.52 (0.87, 2.67)	0.97 (0.51, 1.83)
Model 2	1.00 (Reference)	1.62 (0.92, 2.88)	0.99 (0.52, 1.88)
Model 3	1.00 (Reference)	1.72 (0.94, 3.13)	1.03 (0.53, 2.01)
**SGA**			
Model 1	1.00 (Reference)	1.21 (0.41, 3.53)	1.70 (0.65, 4.46)
Model 2	1.00 (Reference)	1.28 (0.44, 3.75)	1.75 (0.66, 4.63)
Model 3	1.00 (Reference)	1.68 (0.54, 5.17)	2.35 (0.84, 6.60)
**NICU admission**			
Model 1	1.00 (Reference)	0.95 (0.63, 1.43)	0.98 (0.65, 1.47)
Model 2	1.00 (Reference)	1.01 (0.66, 1.53)	1.00 (0.66, 1.52)
Model 3	1.00 (Reference)	1.04 (0.68, 1.60)	1.03 (0.67, 1.59)

Abbreviations: SGA, small for gestational age. OR: odds ratio; CI, confidence interval. Model 1: crude model; Model 2: adjusted for social–economic status, including per capita annual disposable income levels and education degree; Model 3: further adjusted for maternal age at delivery, obesity during pregnancy, whether the conception was planned or not and whether it was achieved through assisted reproductive technology, history of abnormal pregnancy, and whether there was a previous preterm infant.

**Table 4 vaccines-11-01710-t004:** Primary neonatal outcomes in the subgroup analysis according to different vaccine types.

	Unvaccinated	Sinopharm-BIBP	Sinovac-Coronavac
	N = 487	N = 231	N = 138
**Preterm**			
Model 1	1.00 (Reference)	1.50 (0.89, 2.53)	0.82 (0.39, 1.75)
Model 2	1.00 (Reference)	1.55 (0.91, 2.64)	0.87 (0.41, 1.87)
Model 3	1.00 (Reference)	1.64 (0.93, 2.88)	0.89 (0.41, 1.96)
**SGA**			
Model 1	1.00 (Reference)	2.16 (0.92, 5.07)	0.64 (0.14, 2.91)
Model 2	1.00 (Reference)	2.24 (0.95, 5.28)	0.67 (0.15, 3.09)
Model 3	1.00 (Reference)	3.02 (1.20, 7.63)	0.87 (0.18, 4.15)
**NICU admission**			
Model 1	1.00 (Reference)	1.02 (0.70, 1.48)	0.87 (0.55, 1.39)
Model 2	1.00 (Reference)	1.05 (0.72, 1.54)	0.92 (0.57, 1.48)
Model 3	1.00 (Reference)	1.09 (0.73, 1.61)	0.95 (0.58, 1.55)

Abbreviations: SGA, small for gestational age. Model 1: crude model; Model 2: adjusted for social–economic status, including per capita annual disposable income levels and education level; Model 3: further adjusted for maternal age at delivery, obesity during pregnancy, whether the conception was planned or not and whether it was achieved through assisted reproductive technology, history of abnormal pregnancy, and whether there was a previous preterm infant.

## Data Availability

The data generated during the current study are available from the corresponding author upon reasonable request.

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
