# Peer review of "Association of Maternal Inactivated COVID-19 Vaccination within 3 Months before Conception with Neonatal Outcomes"

_vaccines, 2023, doi:10.3390/vaccines11111710_

Round 1
Reviewer 1 Report
Comments and Suggestions for Authors
The manuscript delves into a cohort study executed in a tertiary hospital in China, centering on singleton live births between March 1, 2022, and June 30, 2022. The study's primary objective is to scrutinise the repercussions of maternal COVID-19 vaccination on neonates. The mothers who had received the initial dose of the COVID-19 vaccine before conception were categorized as the vaccinated group, whereas those who remained unvaccinated before or during pregnancy were placed in the unvaccinated group. The study aligns with the Strengthening the Reporting of Observational Studies in Epidemiology (STROBE) reporting guideline, ensuring a high standard of methodological rigor. The analytical framework employed logistic regression to ascertain the odds ratios (ORs) and 95% confidence intervals (CIs) concerning various outcomes while adjusting for factors such as socio-economic status, maternal age at delivery, and obesity during pregnancy, among other pertinent variables. This research revealed no notable disparities between newborns birthed to women who were administered inactivated vaccines before conception and those who were not vaccinated.
Comment
General:
1. The brand of vaccine administered to the patient should be stated. Indicate whether participants receiving vaccines of different brands will affect their statistical results. If there is no relevant information, it should also be written in "limitation".
2. The author should exercise caution when referencing articles, particularly in ensuring that the type of vaccine discussed in the referenced articles aligns with the inactive vaccine mentioned in this paper. Any discrepancy in the type of vaccine could lead to misinterpretation or misrepresentation of the information, which in turn could undermine the accuracy and credibility of the discussions and conclusions drawn in this paper.
3. The discussion on the safety of inactive vaccines is not a new venture; extensive research has been conducted with various other types of vaccines, encompassing the analysis of their effects during pregnancy. While the paper aptly presents a localized analysis of China's vaccination strategy, it fails to advance a novel argument or provide new insights that could galvanise further discussions or investigations in a global context. The essence of the article’s main point tends to traverse already explored territories, hence adding little to the broader scientific discourse on vaccine safety and efficacy.
Minor:
1. The statement in the introduction (Line 71-73), "However, a delay of 2 months is recommended for those undergoing assisted reproduction treatments or experiencing severe immune responses following vaccination, although this recommendation lacks strong evidence," seems ill-suited for inclusion in the article. Given the rapidly evolving nature of the pandemic and the increasing number of vaccinated individuals, this information may already be outdated or no longer applicable. The article must reflect the most current and accurate information, especially when discussing recommendations related to vaccination and reproductive treatments. The lack of strong evidence backing this recommendation further undermines its relevance and accuracy, potentially misleading readers. It would be advisable to either update this information with the latest findings or omit this statement to maintain the integrity and credibility of the article.
2. Line 114-115: None of their mothers ( instead of their mother) were infected by COVID-19 before or during pregnancy: How to confirm that the participants were not infected during pregnancy. Is there any immunological evidence?
3. Table 1 presents the content concerning "Abnormality of umbilical cord". It is crucial to ascertain whether there is literature evidence supporting a correlation between the side effects of the vaccine and this particular abnormality. If no such evidence exists, the rationale behind including this statistical variable should be clearly elucidated in the text.
4. Line 249: "in pro very preterm birth" should be modified
Reviewer 2 Report
Comments and Suggestions for Authors
General Comments:
The manuscript titled "Effects of COVID-19 Vaccination Before Conception" aims to explore the potential impact of COVID-19 vaccination on pregnancy outcomes. The study's relevance, especially given the global emphasis on vaccination, is commendable. However, several aspects require further clarification and elaboration for the paper to achieve its full potential.
Major Comments:
- Line 42-44: The introduction seems to jump abruptly into the topic without providing adequate context. Consider providing a brief background about the general importance of vaccines before delving into specific studies and their results.
- Line 58: It is stated that "the number of cases has increased by 20%," but there's no reference to a baseline or time period. This statement needs clarity. Over what duration has there been a 20% increase?
- Methodology (Lines 78-80): The rationale behind selecting a cross-sectional study design is absent. The authors should elucidate the reasons for this design choice to demonstrate its appropriateness for the research question at hand.
- Participant Details (Lines 85-89): The manuscript lacks comprehensive details about participant recruitment. It is crucial to provide any inclusion/exclusion criteria, demographics, and other relevant characteristics to understand the study sample's context.
- Line 93-95: The methodology employed in the study is introduced rather abruptly. A smooth transition, explaining why this particular method was chosen or its significance, would benefit readers unfamiliar with the technique.
- Statistical Significance (Lines 123-125): The mention of statistical significance in the results section is vague. Including p-values or confidence intervals would give more weight to the claims made.
- Line 127-130: The results section references several tables and figures, but without visual aids in this format, it's challenging to follow. Ensure that all referenced tables and figures are appropriately labelled and included in the final draft.
- Referencing (Throughout the document): The referencing style seems inconsistent. The authors need to ensure that they adhere to the MDPI referencing style. Some references appear incomplete or incorrectly formatted.
- Discussion (Lines 155-157): The discussion section makes comparisons with other regions without specific references. This diminishes the strength of the argument. Providing precise references for these comparative statements is vital.
- Line 180-185: The discussion dives into a comparison with other studies but doesn't provide a clear distinction between the current study's results and others. Consider using a table or a summary to provide a side-by-side comparison for clearer understanding.
- Several researchers have reported that COVID-19 infections/replication was lower when higher temperatures and humidity were seen in the environment. The study was performed in the summer and may be further extended to the winter season, where the COVID-19 infection/replication rates were higher, which can draw the concluding remarks.
- Researchers have mentioned using inactivated COVID-19. As different individuals respond to vaccines differently, so does the immune response, as the authors correctly pointed out that the activation of maternal immune response may potentially influence fetal acceptance and delivery. Serum antibody levels of vaccinated individuals against COVID-19 should also be considered at one or several points after vaccination, as this will affect the study design and results.
- The mothers were previously infected with COVID-19 before receiving the vaccine, which should also be a primary criterion for the study.
- It would be better to plot the graph as in Figure. 2 (representing the vaccinated individuals) for the unvaccinated individuals and compare them with color variation in the same graph, as the entire study is based on the comparison among them.
Minor Comments:
- Formatting: Throughout the document, ensure uniformity in the style, especially in the headings and subheadings.
- Grammar/Typographical Errors: There are a few typographical and grammatical errors scattered throughout the document. A thorough proofreading is recommended.
- Figures (Line 140): If figures are mentioned, ensure they are appropriately embedded or referenced in the final manuscript.
- Line 23: Typographical error: "stuides" should be "studies."
- Line 35: Consider rephrasing "Which leads to the conclusion" to "This suggests that."
- Line 105-106: The phrasing "it can be observed that" is passive. Consider rephrasing for clarity.
- Line 150: There's a missing citation after the statement about vaccine efficacy.
- Line 165: "Whom" should be "who" in the context of this sentence.
- Line 210-211: The conclusion could benefit from a brief summary of the main findings before discussing the implications.
- Line 230: Ensure all references follow a consistent format. There are discrepancies in the way authors' names and publication dates are presented.
- Line 38: Is 3700 number is a fatality?
- Table 1: Paternal is misspelled
Recommendation:
Given the paper's potential importance but also the issues raised, I recommend a "Revise and Resubmit." The authors should address the major and minor comments thoroughly before the manuscript can be considered for publication
Comments on the Quality of English LanguageMinor editing in English is required
Reviewer 3 Report
Comments and Suggestions for Authors
Major issues
-The objectives of the study must be defined and phrased clearly.
-Please indicate clearly the name of the vaccine used in this study. As it is now, this is very abstract and confusing.
-2.1. The inclusion and exclusion criteria are clear, but which were the general selection criteria for people to be included into the study? Please describe.
-Twins / triplets. How many of the infants were twins / triplets? If twins / triplets were included in the study, did you count them as one (same mother) or as two / three (two / three infants) cases? In that case, how did you adjust in the statistical analysis for identical data?
-Did you take into account, cases of covid-19 infection of the pregnant women during gestation, after inclusion in the study? How did you deal with such cases? Also, same questions for infantile covid-19 infections. Please explain and discuss.
Minor issues
-The second paragraph of the Introduction is long and includes well-known facts. Hence, I suggest to reduce it in length by deleting superfluous sentences.
-Statistical analysis: please mention the test employed to determine normal distribution of data.
-Discussion can be better divided into two parts for better flow of reading.
Overall
Useful study that can be published after revision and clarifications as indicated above.
Reviewer 4 Report
Comments and Suggestions for Authors
Estimated Authors,
first of all, thank you for the opportunity of reviewing this interesting study on the gestational outcome in children vaccinated vs. not vaccinated against SARS-CoV-2 in mainland China.
The present article is of particular interest as, while the evolution of the pandemic reasonably reduces the need for massive vaccination campaigns, the possible vaccination of pregnant women will remain in the next years.
According to this study, no significant effect of SARS-CoV-2 vaccine could be identified in pregnancies, and therefore we could agree that women can be safely vaccinated against this pathogen.
On the other hand, the quality of the present paper is (at the moment) formally inappropriate for guaranteeing its acceptance as it is.
Several adjustements must but hopefully could be done (in a relatively short timeframe). Most notably:
1) provide more information about the vaccine delivered to the vaccinated women: please provide the kind of vaccine (e.g. sinofarm? valneva? please explain);
2) please provide, in both main text and individual tables the meaning of the acronyms and translate in € or $ the cut off for annual income of recruited women;
3) Table 3+4: please consider (optional recommendation) the alternative implementation of a forrest plot instead of a table
4) The main text is mostly correct from an English language point of view, but the section from row 106 to row 120 is quite unclear in phrasing and reporting. Please double check.
As you can see, the whole of my recommendations are from a formal point of view, therefore I'm confident that you could quickly cope with the required adjustements.
Comments on the Quality of English LanguageThe main text is mostly correct from an English language point of view, but the section from row 106 to row 120 is quite unclear in phrasing and reporting. Please double check.
Round 2
Reviewer 1 Report
Comments and Suggestions for Authors
First, I would like to highlight that the primary concept of this manuscript appears to have significant overlap with a previously published article titled "Maternal and neonatal safety of COVID-19 vaccination during the peri-pregnancy period: A prospective study" in the Journal of Medical Virology, 2022.
Given this prior publication, authors must provide a compelling argument distinguishing their work from the aforementioned study. The scientific community and readers of Vaccines would benefit from understanding the unique contributions of your research and why it warrants publication despite the similarities.
Second, I want to draw attention to the section between Lines 380-390. The discussion in this segment appears to deviate from the central theme of the manuscript. The references cited in this section predominantly address the safety of COVID-19 vaccine administration during pregnancy. However, the primary focus of the manuscript is on the administration of inactive COVID-19 vaccines "before" conception. Given the manuscript's emphasis on the potential impacts of inactive COVID-19 vaccine administration BEFOREpregnancy, it would be beneficial for the authors to provide a more in-depth discussion of the scientific evidence regarding whether such vaccination would influence neonatal health. While informative, the current discussion in Lines 380-390 may not directly align with the manuscript's core objective.
I recommend that the authors revisit this section and ensure the discussion is more closely tied to the manuscript's main theme. Expanding on the scientific evidence related to the effects of inactive vaccine administration before pregnancy on neonatal health would enhance the manuscript's coherence and relevance.
Third, the authors have defined the study group as individuals who received the vaccine within three months before conception. The authors should explain why they chose the three-month timeframe specifically. From a reproductive medicine perspective, various physiological and hormonal changes occur in preparation for conception, and the impact of external factors, such as vaccines, can vary depending on the timing of administration relative to conception.
Fourth, several variables used in your statistical analyses are not adequately defined or explained in terms of their significance to the study's objectives. Specifically, the variable "Paternal vaccination" caught my attention. It would be beneficial for readers to understand why and how paternal vaccination might influence the study's primary outcomes. Please provide a rationale or background on including this variable and its potential implications. Using "US 380" as a cutoff level for assessing household income raises questions. Firstly, the rationale behind this specific amount should be elucidated. Is this based on a recognized poverty line, median income, or another metric? Secondly, it's crucial to address the variability in income levels across different cities or regions. Please provide context on how this cutoff was determined and its relevance to the study population.
Fifth, under Table 1, specifically the definition of "History of abnormal pregnancy."
The description states: "a : History of abnormal pregnancy is identified as history of embryonic resorption or fetal abortion such as ectopic gestation, biochemical pregnancy." I would like to point out that classifying an ectopic pregnancy under the categories of embryonic resorption or fetal abortion may not be appropriate. Please
revise the definition.
Finally, while I acknowledge your mention of the rigorous policies in place in your country for detecting COVID-19, it is essential to recognize that many infections can be asymptomatic or present with mild symptoms. Additionally, there might be instances where individuals do not report their symptoms to health authorities. Given this context, the statement -- "The collection COVID-19 infection information was conducted by questionnaire, and it was 'confirmed' that none of their mothers had been infected with COVID-19 either before or during pregnancy" -- appears to be overly definitive. The authors should redefine the” Natural Infection”- It would be prudent to define 'natural infection' based on universally accepted scientific criteria rather than solely relying on participant questionnaires. If possible, consider using the presence of anti-N antibodies for SARS-CoV-2 as a more objective criterion for past infections. If no such tests were performed, it is crucial to acknowledge this as a study limitation.
Comments on the Quality of English Language
There are several instances where the grammar and sentence structure could benefit from revisions to enhance clarity and readability. Proper grammar and clear language are essential for conveying your research effectively and ensuring that the manuscript meets the publication standards of Vaccines
Reviewer 2 Report
Comments and Suggestions for Authors
The authors have addressed all my concerns
Author Response
Comments and Suggestions for Authors
The authors have addressed all my concerns
Response: Thanks again for taking the time to review this manuscript.
Reviewer 3 Report
Comments and Suggestions for Authors
The authors have made all changes required and have improved the manuscript.
Before definite acceptance, I suggest to add a final paragraph in the text to explain how their findings can fit into the official recommendations for vaccination schedules against covid-19. This will be very helpful for clinicians across the world.
Author Response
Comments and Suggestions for Authors
The authors have made all changes required and have improved the manuscript.
Before definite acceptance, I suggest to add a final paragraph in the text to explain how their findings can fit into the official recommendations for vaccination schedules against covid-19. This will be very helpful for clinicians across the world.
Response: We do really appreciate your comments. We have already added some content in final paragraph as follows:
“...Thus, our data supports implementing COVID-19 vaccinations for women preparing for pregnancy. Additionally, it suggests that inactivated COVID-19 vaccination admin-istered within 3 months before conception appears to be safe for newborns...”